# Mirrored Langevin Dynamics

Ya-Ping Hsieh      Ali Kavis      Paul Rolland      Volkan Cevher

Laboratory for Information and Inference Systems (LIONS),
EPFL, Lausanne, Switzerland
{ya-ping.hsieh, ali.kavis, paul.rolland, volkan.cevher}@epfl.ch

## Abstract

We consider the problem of sampling from *constrained* distributions, which has posed significant challenges to both non-asymptotic analysis and algorithmic design. We propose a unified framework, which is inspired by the classical mirror descent, to derive novel first-order sampling schemes. We prove that, for a general target distribution with strongly convex potential, our framework implies the existence of a first-order algorithm achieving $\tilde{O}(\epsilon^{-2}d)$ convergence, suggesting that the state-of-the-art $\tilde{O}(\epsilon^{-6}d^5)$ can be vastly improved. With the important Latent Dirichlet Allocation (LDA) application in mind, we specialize our algorithm to sample from Dirichlet posteriors, and derive the first non-asymptotic $\tilde{O}(\epsilon^{-2}d^2)$ rate for first-order sampling. We further extend our framework to the mini-batch setting and prove convergence rates when only stochastic gradients are available. Finally, we report promising experimental results for LDA on real datasets.

## 1   Introduction

Many modern learning tasks involve sampling from a high-dimensional and large-scale distribution, which calls for algorithms that are scalable with respect to both the dimension and the data size. One approach [32] that has found wide success is to discretize the **Langevin Dynamics**:

$$d\mathbf{X}_t = -\nabla V(\mathbf{X}_t)dt + \sqrt{2}d\mathbf{B}_t, \tag{1.1}$$

where $e^{-V(\mathbf{x})}d\mathbf{x}$ presents a target distribution and $\mathbf{B}_t$ is a $d$-dimensional Brownian motion. Such a framework has inspired numerous first-order sampling algorithms [1, 7, 13, 15, 18, 19, 26, 29], and the convergence rates are by now well-understood for unconstrained and log-concave distributions [8, 12, 14].

However, applying (1.1) to sampling from *constrained* distributions (i.e., when $V$ has a bounded convex domain) remains a difficult challenge. From the theoretical perspective, there are only two existing algorithms [4, 5] that possess non-asymptotic guarantees, and their rates are significantly worse than the unconstrained scenario under the same assumtions; *cf.*, Table 1. Furthermore, many important constrained distributions are inherently non-log-concave. A prominent instance is the **Dirichlet posterior**, which, in spite of the presence of several tailor-made first-order algorithms [18, 26], is still lacking a non-asymptotic guarantee.

In this paper, we aim to bridge these two gaps at the same time. For general constrained distributions with a strongly convex potential $V$, we prove the existence of a first-order algorithm that achieves the same convergence rates as if there is no constraint at all, suggesting the state-of-the-art $\tilde{O}(\epsilon^{-6}d^5)$ can be brought down to $\tilde{O}(\epsilon^{-2}d)$. When specialized to the important case of simplex constraint, we provide the first non-asymptotic guarantee for

Dirichlet posteriors, $\tilde{O}(\epsilon^{-2}d^2 R_0)$ for deterministic and $\tilde{O}\left(\epsilon^{-2}(Nd + \sigma^2)R_0\right)$ for the stochastic version of our algorithms; *cf.*, **Example 1** and **2** for the involved parameters.

Our framework combines ideas from the **Mirror Descent** [2, 25] algorithm for optimization and the theory of Optimal Transport [31]. Concretely, for constrained sampling problems, we propose to use the *mirror map* to transform the target into an unconstrained distribution, whereby many existing methods apply. Optimal Transport theory then comes in handy to relate the convergence rates between the original and transformed problems. For simplex constraints, we use the entropic mirror map to design practical first-order algorithms that possess rigorous guarantees, and are amenable to mini-batch extensions.

The rest of the paper is organized as follows. We briefly review the notion of push-forward measures in Section 2. In Section 3, we propose the **Mirrored Langevin Dynamics** and prove its convergence rates for constrained sampling problems. Mini-batch extensions are derived in Section 4. Finally, in Section 5, we provide synthetic and real-world experiments to demonstrate the empirical efficiency of our algorithms.

## 1.1 Related Work

**First-Order Sampling Schemes with Langevin Dynamics:** There exists a bulk of literature on (stochastic) first-order sampling schemes derived from Langevin Dynamics or its variants [1, 4–6, 8, 9, 12, 14, 16, 20, 26, 32]. However, to our knowledge, this work is the first to consider mirror descent extensions of the Langevin Dynamics.

The authors in [21] proposed a formalism that can, in principle, incorporate any variant of Langevin Dynamics for a given distribution $e^{-V(\mathbf{x})}\mathrm{d}\mathbf{x}$. The Mirrored Langevin Dynamics, however, is targeting the push-forward measure $e^{-W(\mathbf{y})}\mathrm{d}\mathbf{y}$ (see Section 3.1), and hence our framework is not covered in [21].

For Dirichlet posteriors, there is a similar variable transformation as our entropic mirror map in [26] (see the "reduced-natural parametrization" therein). The dynamics in [26] is nonetheless drastically different from ours, as there is a position-dependent matrix multiplying the Brownian motion, whereas our dynamics has no such feature; see (3.2).

**Mirror Descent-Type Dynamics for Stochastic Optimization:** Although there are some existing work on mirror descent-type dynamics for *stochastic optimization* [17, 24, 27, 33], we are unaware of any prior result on sampling.

## 2 Preliminaries

### 2.1 Notation

In this paper, all Lipschitzness and strong convexity are with respect to the Euclidean norm $\|\cdot\|$. We use $\mathcal{C}^k$ to denote $k$-times differentiable functions with continuous $k^{\mathrm{th}}$ derivative. The Fenchel dual [28] of a function $h$ is denoted by $h^\star$. Given two mappings $T, F$ of proper dimensions, we denote their composite map by $T \circ F$. For a probability measure $\mu$, we write $\mathbf{X} \sim \mu$ to mean that "$\mathbf{X}$ is a random variable whose probability law is $\mu$".

### 2.2 Push-Forward and Optimal Transport

Let $\mathrm{d}\mu = e^{-V(\mathbf{x})}\mathrm{d}\mathbf{x}$ be a probability measure with support $\mathcal{X} := \mathrm{dom}(V) = \{\mathbf{x} \in \mathbb{R}^d \mid V(\mathbf{x}) < +\infty\}$, and $h$ be a convex function on $\mathcal{X}$. Throughout the paper we assume:

**Assumption 1.** *$h$ is closed, proper, $h \in \mathcal{C}^2$, and $\nabla^2 h \succ 0$ on $\mathcal{X} \subset \mathbb{R}^d$.*

**Assumption 2.** *All measures have finite second moments.*

**Assumption 3.** *All measures vanish on sets with Hausdorff dimension [22] at most $d-1$.*

The gradient map $\nabla h$ induces a new probability measure $\mathrm{d}\nu := e^{-W(\mathbf{y})}\mathrm{d}\mathbf{y}$ through $\nu(E) = \mu\left(\nabla h^{-1}(E)\right)$ for every Borel set $E$ on $\mathbb{R}^d$. We say that $\nu$ is the **push-forward measure** of $\mu$ under $\nabla h$, and we denote it by $\nabla h \# \mu = \nu$. If $\mathbf{X} \sim \mu$ and $\mathbf{Y} \sim \nu$, we will sometimes abuse the notation by writing $\nabla h \# \mathbf{X} = \mathbf{Y}$ to mean $\nabla h \# \mu = \nu$.

If $\nabla h \# \mu = \nu$, the triplet $(\mu, \nu, h)$ must satisfy the Monge-Ampère equation:

$$e^{-V} = e^{-W \circ \nabla h} \det \nabla^2 h. \tag{2.1}$$

Using $(\nabla h)^{-1} = \nabla h^{\star}$ and $\nabla^2 h \circ \nabla h^{\star} = \nabla^2 h^{\star -1}$, we see that (2.1) is equivalent to

$$e^{-W} = e^{-V \circ \nabla h^{\star}} \det \nabla^2 h^{\star} \tag{2.2}$$

which implies $\nabla h^{\star} \# \nu = \mu$.

The 2-Wasserstein distance between $\mu_1$ and $\mu_2$ is defined by[1]

$$\mathcal{W}_2^2(\mu_1, \mu_2) := \inf_{T: T \# \mu_1 = \mu_2} \int \|\mathbf{x} - T(\mathbf{x})\|^2 \mathrm{d}\mu_1(\mathbf{x}). \tag{2.3}$$

# 3 Mirrored Langevin Dynamics

This section demonstrates a framework for transforming constrained sampling problems into unconstrained ones. We then focus on applications to sampling from strongly log-concave distributions and simplex-constrained distributions, even though the framework is more general and future-proof.

## 3.1 Motivation and Algorithm

We begin by briefly recalling the mirror descent (MD) algorithm for optimization. In order to minimize a function over a bounded domain, say $\min_{\mathbf{x} \in \mathcal{X}} f(\mathbf{x})$, MD uses a mirror map $h$ to transform the primal variable $\mathbf{x}$ into the dual space $\mathbf{y} := \nabla h(\mathbf{x})$, and then performs gradient updates in the dual: $\mathbf{y}^+ = \mathbf{y} - \beta \nabla f(\mathbf{x})$ for some step-size $\beta$. The mirror map $h$ is chosen to adapt to the geometry of the constraint $\mathcal{X}$, which can often lead to faster convergence [25] or, more pivotal to this work, an **unconstrained** optimization problem [2].

Inspired by the MD framework, we would like to use the mirror map idea to remove the constraint for sampling problems. Toward this end, we first establish a simple fact [30]:

**Theorem 1.** *Let $h$ satisfy **Assumption 1**. Suppose that $\mathbf{X} \sim \mu$ and $\mathbf{Y} = \nabla h(\mathbf{X})$. Then $\mathbf{Y} \sim \nu := \nabla h \# \mu$ and $\nabla h^{\star}(\mathbf{Y}) \sim \mu$.*

*Proof.* For any Borel set $E$, we have $\nu(E) = \mathbb{P}(\mathbf{Y} \in E) = \mathbb{P}(\mathbf{X} \in \nabla h^{-1}(E)) = \mu(\nabla h^{-1}(E))$. Since $\nabla h$ is one-to-one, $\mathbf{Y} = \nabla h(\mathbf{X})$ if and only if $\mathbf{X} = \nabla h^{-1}(\mathbf{Y}) = \nabla h^{\star}(\mathbf{Y})$. $\quad\square$

In the context of sampling, **Theorem 1** suggests the following simple procedure: For any target distribution $e^{-V(\mathbf{x})} \mathrm{d}\mathbf{x}$ with support $\mathcal{X}$, we choose a mirror map $h$ on $\mathcal{X}$ satisfying **Assumption 1**, and we consider the **dual distribution** associated with $e^{-V(\mathbf{x})} \mathrm{d}\mathbf{x}$ and $h$:

$$e^{-W(\mathbf{y})} \mathrm{d}\mathbf{y} := \nabla h \# e^{-V(\mathbf{x})} \mathrm{d}\mathbf{x}. \tag{3.1}$$

**Theorem 1** dictates that if we are able to draw a sample $\mathbf{Y}$ from $e^{-W(\mathbf{y})} \mathrm{d}\mathbf{y}$, then $\nabla h^{\star}(\mathbf{Y})$ immediately gives a sample for the desired distribution $e^{-V(\mathbf{x})} \mathrm{d}\mathbf{x}$. Furthermore, suppose for the moment that $\mathrm{dom}(h^{\star}) = \mathbb{R}^d$, so that $e^{-W(\mathbf{y})} \mathrm{d}\mathbf{y}$ is unconstrained. Then we can simply exploit the classical Langevin Dynamics (1.1) to efficiently take samples from $e^{-W(\mathbf{y})} \mathrm{d}\mathbf{y}$.

The above reasoning leads us to set up the **Mirrored Langevin Dynamics** (MLD):

$$\mathbf{MLD} \equiv \begin{cases} \mathrm{d}\mathbf{Y}_t = -(\nabla W \circ \nabla h)(\mathbf{X}_t)\mathrm{d}t + \sqrt{2}\mathrm{d}\mathbf{B}_t \\ \mathbf{X}_t = \nabla h^{\star}(\mathbf{Y}_t) \end{cases}. \tag{3.2}$$

Notice that the stationary distribution of $\mathbf{Y}_t$ in MLD is $e^{-W(\mathbf{y})} \mathrm{d}\mathbf{y}$, since $\mathrm{d}\mathbf{Y}_t$ is nothing but the Langevin Dynamics (1.1) with $\nabla V \leftarrow \nabla W$. As a result, we have $\mathbf{X}_t \to \mathbf{X}_\infty \sim e^{-V(\mathbf{x})} \mathrm{d}\mathbf{x}$.

| Assumption | $D(\cdot\|\cdot)$ | $\mathcal{W}_2$ | $d_{\mathrm{TV}}$ | Algorithm |
|---|---|---|---|---|
| $LI \succeq \nabla^2 V \succeq mI$ | unknown | unknown | $\tilde{O}\left(\epsilon^{-6}d^5\right)$ | MYULA [4] |
| $LI \succeq \nabla^2 V \succeq 0$ | unknown | unknown | $\tilde{O}\left(\epsilon^{-12}d^{12}\right)$ | PLMC [5] |
| $\nabla^2 V \succeq mI$ | $\tilde{O}\left(\epsilon^{-1}d\right)$ | $\tilde{O}\left(\epsilon^{-2}d\right)$ | $\tilde{O}\left(\epsilon^{-2}d\right)$ | MLD; this work |
| $LI \succeq \nabla^2 V \succeq mI,$ <br> $V$ unconstrained | $\tilde{O}\left(\epsilon^{-1}d\right)$ | $\tilde{O}\left(\epsilon^{-2}d\right)$ | $\tilde{O}\left(\epsilon^{-2}d\right)$ | Langevin Dynamics [8, 11, 14] |

Table 1: Convergence rates for sampling from $e^{-V(\mathbf{x})}\mathrm{d}\mathbf{x}$ with $\mathrm{dom}(V)$ bounded

Using (2.1), we can equivalently write the $\mathrm{d}\mathbf{Y}_t$ term in (3.2) as

$$\mathrm{d}\mathbf{Y}_t = -\nabla^2 h(\mathbf{X}_t)^{-1}\Big(\nabla V(\mathbf{X}_t) + \nabla\log\det\nabla^2 h(\mathbf{X}_t)\Big)\mathrm{d}t + \sqrt{2}\mathrm{d}\mathbf{B}_t.$$

In order to arrive at a practical algorithm, we then discretize the MLD, giving rise to the following equivalent iterations:

$$\mathbf{y}^{t+1} - \mathbf{y}^t = \begin{cases} -\beta^t\nabla W(\mathbf{y}^t) + \sqrt{2\beta^t}\boldsymbol{\xi}^t \\ -\beta^t\nabla^2 h(\mathbf{x}^t)^{-1}\Big(\nabla V(\mathbf{x}^t) + \nabla\log\det\nabla^2 h(\mathbf{x}^t)\Big) + \sqrt{2\beta^t}\boldsymbol{\xi}^t \end{cases} \quad (3.3)$$

where in both cases $\mathbf{x}^{t+1} = \nabla h^\star(\mathbf{y}^{t+1})$, $\boldsymbol{\xi}^t$'s are i.i.d. standard Gaussian, and $\beta^t$'s are step-sizes. The first formulation in (3.3) is useful when $\nabla W$ has a tractable form, while the second one can be computed using solely the information of $V$ and $h$.

Next, we turn to the convergence of discretized MLD. Since $\mathrm{d}\mathbf{Y}_t$ in (3.2) is the classical Langevin Dynamics, and since we have assumed that $W$ is unconstrained, it is typically not difficult to prove the convergence of $\mathbf{y}^t$ to $\mathbf{Y}_\infty \sim e^{-W(\mathbf{y})}\mathrm{d}\mathbf{y}$. However, what we ultimately care about is the guarantee on the primal distribution $e^{-V(\mathbf{x})}\mathrm{d}\mathbf{x}$. The purpose of the next theorem is to fill the gap between primal and dual convergence.

We consider three most common metrics in evaluating approximate sampling schemes, namely the 2-Wasserstein distance $\mathcal{W}_2$, the total variation $d_{\mathrm{TV}}$, and the relative entropy $D(\cdot\|\cdot)$.

**Theorem 2** (Convergence in $\mathbf{y}^t$ implies convergence in $\mathbf{x}^t$). *For any $h$ satisfying **Assumption 1**, we have $d_{\mathrm{TV}}(\nabla h\#\mu_1, \nabla h\#\mu_2) = d_{\mathrm{TV}}(\mu_1, \mu_2)$ and $D(\nabla h\#\mu_1\|\nabla h\#\mu_2) = D(\mu_1\|\mu_2)$. In particular, we have $d_{\mathrm{TV}}(\mathbf{y}^t, \mathbf{Y}_\infty) = d_{\mathrm{TV}}(\mathbf{x}^t, \mathbf{X}_\infty)$ and $D(\mathbf{y}^t\|\mathbf{Y}_\infty) = D(\mathbf{x}^t\|\mathbf{X}_\infty)$ in (3.3).*

*If, furthermore, $h$ is $\rho$-strongly convex: $\nabla^2 h \succeq \rho I$. Then $\mathcal{W}_2(\mathbf{x}^t, \mathbf{X}_\infty) \leq \frac{1}{\rho}\mathcal{W}_2(\mathbf{y}^t, \mathbf{Y}_\infty)$.*

*Proof.* See **Appendix A**. $\qquad\square$

### 3.2 Applications to Sampling from Constrained Distributions

We now consider applications of MLD. For strongly log-concave distributions with general constraint, we prove matching rates to that of unconstrained ones; see Section 3.2.1. In Section 3.2.2, we consider the important case where the constraint is a probability simplex[2].

#### 3.2.1 Sampling from a strongly log-concave distribution with constraint

As alluded to in the introduction, the existing convergence rates for constrained distributions are significantly worse than their unconstrained counterparts; see Table 1 for a comparison. The main result of this subsection is the existence of a "good" mirror map for *arbitrary* constraint, with which the dual distribution $e^{-W(\mathbf{y})}\mathrm{d}\mathbf{y}$ becomes unconstrained:

**Theorem 3** (Existence of a good mirror map for MLD). *Let $\mathrm{d}\mu(\mathbf{x}) = e^{-V(\mathbf{x})}\mathrm{d}\mathbf{x}$ be a probability measure with bounded convex support such that $V \in \mathcal{C}^2$, $\nabla^2 V \succeq mI \succ 0$, and $V$ is bounded away from $+\infty$ in the interior of the support. Then there exists a mirror map $h \in \mathcal{C}^2$ such that the discretized MLD (3.3) yields*

$$D\left(\mathbf{x}^T\|\mathbf{X}_\infty\right) = \tilde{O}\left(\frac{d}{T}\right), \quad \mathcal{W}_2\left(\mathbf{x}^T, \mathbf{X}_\infty\right) = \tilde{O}\left(\sqrt{\frac{d}{T}}\right), \quad d_{\mathrm{TV}}\left(\mathbf{x}^T, \mathbf{X}_\infty\right) = \tilde{O}\left(\sqrt{\frac{d}{T}}\right).$$

*Proof.* See **Appendix C.** □

**Remark 1.** *We remark that **Theorem 3** is only an existential result, not an actual algorithm. Practical algorithms are considered in the next subsection.*

### 3.2.2 Sampling Algorithms on Simplex

We apply the discretized MLD (3.3) to the task of sampling from distributions on the probability simplex $\Delta_d := \{\mathbf{x} \in \mathbb{R}^d \mid \sum_{i=1}^d x_i \leq 1, x_i \geq 0\}$, which is instrumental in many fields of machine learning and statistics.

On a simplex, the most natural choice of $h$ is the entropic mirror map [2], which is well-known to be 1-strongly convex:

$$h(\mathbf{x}) = \sum_{\ell=1}^d x_i \log x_\ell + \left(1 - \sum_{\ell=1}^d x_\ell\right) \log\left(1 - \sum_{\ell=1}^d x_\ell\right), \text{ where } 0 \log 0 := 0. \quad (3.4)$$

In this case, the associated dual distribution can be computed explicitly.

**Lemma 1** (Sampling on a simplex with entropic mirror map). *Let $e^{-V(\mathbf{x})}\mathrm{d}\mathbf{x}$ be the target distribution on $\Delta_d$, $h$ be the entropic mirror map (3.4), and $e^{-W(\mathbf{y})}\mathrm{d}\mathbf{y} := \nabla h \# e^{-V(\mathbf{x})}\mathrm{d}\mathbf{x}$. Then the potential $W$ of the push-forward measure admits the expression*

$$W(\mathbf{y}) = V \circ \nabla h^\star(\mathbf{y}) - \sum_{\ell=1}^d y_\ell + (d+1)h^\star(\mathbf{y}) \quad (3.5)$$

*where $h^\star(\mathbf{y}) = \log\left(1 + \sum_{\ell=1}^d e^{y_\ell}\right)$ is the Fenchel dual of $h$, which is strictly convex and 1-Lipschitz gradient.*

*Proof.* See **Appendix D.** □

Crucially, we have $\mathrm{dom}(h^\star) = \mathbb{R}^d$, so that the Langevin Dynamics for $e^{-W(\mathbf{y})}\mathrm{d}\mathbf{y}$ is **unconstrained**.

Based on **Lemma 1**, we now present the surprising case of the *non-log-concave* Dirichlet posteriors, a distribution of central importance in topic modeling [3], for which the dual distribution $e^{-W(\mathbf{y})}\mathrm{d}\mathbf{y}$ becomes strictly *log-concave*.

**Example 1** (Dirichlet Posteriors). Given parameters $\alpha_1, \alpha_2, ..., \alpha_{d+1} > 0$ and observations $n_1, n_2, ..., n_{d+1}$ where $n_\ell$ is the number of appearance of category $\ell$, the probability density function of the Dirichlet posterior is

$$p(\mathbf{x}) = \frac{1}{C} \prod_{\ell=1}^{d+1} x_\ell^{n_\ell + \alpha_\ell - 1}, \quad \mathbf{x} \in \mathrm{int}\,(\Delta_d) \quad (3.6)$$

where $C$ is a normalizing constant and $x_{d+1} := 1 - \sum_{\ell=1}^d x_\ell$. The corresponding $V$ is

$$V(\mathbf{x}) = -\log p(\mathbf{x}) = \log C - \sum_{\ell=1}^{d+1}(n_\ell + \alpha_\ell - 1)\log x_\ell, \quad \mathbf{x} \in \mathrm{int}\,(\Delta_d).$$

The interesting regime of the Dirichlet posterior is when it is **sparse**, meaning the majority of the $n_\ell$'s are zero and a few $n_k$'s are large, say of order $O(d)$. It is also common to set $\alpha_\ell < 1$ for all $\ell$ in practice. Evidently, $V$ is neither convex nor concave in this case, and no existing non-asymptotic rate can be applied. However, plugging $V$ into (3.5) gives

$$W(\mathbf{y}) = \log C - \sum_{\ell=1}^d (n_\ell + \alpha_\ell)y_\ell + \left(\sum_{\ell=1}^{d+1}(n_\ell + \alpha_\ell)\right)h^\star(\mathbf{y}) \quad (3.7)$$

which, magically, becomes strictly convex and $O(d)$-Lipschitz gradient **no matter what the observations and parameters are!** In view of **Theorem 2** and [14, **Corollary 7**], one can then apply (3.3) to obtain an $\tilde{O}\left(\epsilon^{-2}d^2 R_0\right)$ convergence in relative entropy, where $R_0 := \mathcal{W}_2^2(\mathbf{y}^0, e^{-W(\mathbf{y})}\mathrm{d}\mathbf{y})$ is the initial Wasserstein distance to the target. □

---

**Algorithm 1** Stochastic Mirrored Langevin Dynamics (SMLD)

---

**Require:** Target distribution $e^{-V(\mathbf{x})}\mathrm{d}\mathbf{x}$ where $V = \sum_{i=1}^{N} V_i$, step-sizes $\beta^t$, batch-size $b$
 1: Find $W_i$ such that $e^{-NW_i} \propto \nabla h \# e^{-NV_i}$ for all $i$.
 2: **for** $t \leftarrow 0, 1, \cdots, T-1$ **do**
 3:     Pick a mini-batch $B$ of size $b$ uniformly at random.
 4:     Update $\mathbf{y}^{t+1} = \mathbf{y}^t - \frac{\beta^t N}{b} \sum_{i \in B} \nabla W_i(\mathbf{y}^t) + \sqrt{2\beta^t}\boldsymbol{\xi}^t$
 5:     $\mathbf{x}^{t+1} = \nabla h^\star(\mathbf{y}^{t+1})$          ▷ Update only when necessary.
 6: **end for**
**return** $\mathbf{x}^T$

---

## 4  Stochastic Mirrored Langevin Dynamics

We have thus far only considered **deterministic** methods based on exact gradients. In practice, however, evaluating gradients typically involves one pass over the full data, which can be time-consuming in large-scale applications. In this section, we turn attention to the **mini-batch** setting, where one can use a small subset of data to form stochastic gradients.

Toward this end, we assume:

**Assumption 4** (Primal Decomposibility). *The target distribution $e^{-V(\mathbf{x})}\mathrm{d}\mathbf{x}$ admits a decomposable structure $V = \sum_{i=1}^{N} V_i$ for some functions $V_i$.*

Consider the following common scheme in obtaining stochastic gradients. Given a batch-size $b$, we randomly pick a mini-batch $B$ from $\{1, 2, \ldots, N\}$ with $|B| = b$, and form an unbiased estimate of $\nabla V$ by computing

$$\tilde{\nabla}V \coloneqq \frac{N}{b} \sum_{i \in B} \nabla V_i. \tag{4.1}$$

The following lemma asserts that exactly the same procedure can be carried out in the dual.

**Lemma 2.** *Assume that $h$ is 1-strongly convex. For $i = 1, 2, ..., N$, let $W_i$ be such that*

$$e^{-NW_i} = \nabla h \# \frac{e^{-NV_i}}{\int e^{-NV_i}}. \tag{4.2}$$

*Define $W \coloneqq \sum_{i=1}^{N} W_i$ and $\tilde{\nabla}W \coloneqq \frac{N}{b} \sum_{i \in B} \nabla W_i$, where $B$ is chosen as in (4.1). Then:*
 1. *Primal decomposibility implies dual decomposability: There is a constant $C$ such that $e^{-(W+C)} = \nabla h \# e^{-V}$.*
 2. *For each $i$, the gradient $\nabla W_i$ depends only on $\nabla V_i$ and the mirror map $h$.*
 3. *The gradient estimate is unbiased: $\mathbb{E}\tilde{\nabla}W = \nabla W$.*
 4. *The dual stochastic gradient is more accurate: $\mathbb{E}\|\tilde{\nabla}W - \nabla W\|^2 \leq \mathbb{E}\|\tilde{\nabla}V - \nabla V\|^2$.*

*Proof.* See **Appendix E**.  □

**Lemma 2** furnishes a template for the mini-batch extension of MLD. The pseudocode is detailed in **Algorithm 1**, whose convergence rate is given by the next theorem.

**Theorem 4.** *Let $e^{-V(\mathbf{x})}\mathrm{d}\mathbf{x}$ be a distribution satisfying **Assumption 4**, and $h$ a 1-strongly convex mirror map. Let $\sigma^2 \coloneqq \mathbb{E}\|\tilde{\nabla}V - \nabla V\|^2$ be the variance of the stochastic gradient of $V$ in (4.1). Suppose that the corresponding dual distribution $e^{-W(\mathbf{y})}\mathrm{d}\mathbf{y} = \nabla h \# e^{-V(\mathbf{x})}\mathrm{d}\mathbf{x}$ satisfies $LI \succeq \nabla^2 W \succeq 0$. Then, applying SMLD with constant step-size $\beta^t = \beta$ yields[3]:*

$$D\left(\mathbf{x}^T \| e^{-V(\mathbf{x})}\mathrm{d}\mathbf{x}\right) \leq \sqrt{\frac{2\mathcal{W}_2^2\left(\mathbf{y}^0, e^{-W(\mathbf{y})}\mathrm{d}\mathbf{y}\right)(Ld + \sigma^2)}{T}} = O\left(\sqrt{\frac{Ld + \sigma^2}{T}}\right), \tag{4.3}$$

*provided that $\beta \leq \min\left\{\left[2T\mathcal{W}_2^2\left(\mathbf{y}^0, e^{-W(\mathbf{y})}\mathrm{d}\mathbf{y}\right)(Ld + \sigma^2)\right]^{-\frac{1}{2}}, \frac{1}{L}\right\}$.*

*Proof.* See **Appendix F**. □

**Example 2** (SMLD for Dirichlet Posteriors)**.** For the case of Dirichlet posteriors, we have seen in (3.7) that the corresponding dual distribution satisfies $(N + \Gamma)I \succeq \nabla^2 W \succ 0$, where $N := \sum_{\ell=1}^{d+1} n_\ell$ and $\Gamma := \sum_{\ell=1}^{d+1} \alpha_\ell$. Furthermore, it is easy to see that the stochastic gradient $\tilde{\nabla} W$ can be efficiently computed (see **Appendix G**):

$$\tilde{\nabla} W(\mathbf{y})_\ell := \frac{N}{b} \sum_{i \in B} \nabla W_i(\mathbf{y})_\ell = - \left( \frac{N m_\ell}{b} + \alpha_\ell \right) + (N + \Gamma) \frac{e^{y_\ell}}{1 + \sum_{k=1}^d e^{y_k}}, \qquad (4.4)$$

where $m_\ell$ is the number of observations of category $\ell$ in the mini-batch $B$. As a result, **Theorem 4** states that SMLD achieves

$$D \left( \mathbf{x}^T \| e^{-V(\mathbf{x})} d\mathbf{x} \right) \leq \sqrt{\frac{2\mathcal{W}_2^2 \left( \mathbf{y}^0, e^{-W(\mathbf{y})} d\mathbf{y} \right) \left( (N + \Gamma)(d+1) + \sigma^2 \right)}{T}} = O \left( \sqrt{\frac{(N + \Gamma)d + \sigma^2}{T}} \right)$$

with a constant step-size. □

# 5  Experiments

We conduct experiments with a two-fold purpose. First, we use a low-dimensional synthetic data, where we can evaluate the total variation error by comparing histograms, to verify the convergence rates in our theory. Second, We demonstrate that the SMLD, modulo a necessary modification for resolving numerical issues, outperforms state-of-the-art first-order methods on the Latent Dirichlet Allocation (LDA) application with Wikipedia corpus.

## 5.1  Synthetic Experiment for Dirichlet Posterior

We implement the deterministic MLD for sampling from an 11-dimensional Dirichlet posterior (3.6) with $n_1 = 10,000$, $n_2 = n_3 = 10$, and $n_4 = n_5 = \cdots = n_{11} = 0$, which aims to capture the sparse nature of real observations in topic modeling. We set $\alpha_\ell = 0.1$ for all $\ell$.

As a baseline comparison, we include the Stochastic Gradient Riemannian Langevin Dynamics (SGRLD) [26] with the expanded-mean parametrization. SGRLD is a tailor-made first-order scheme for simplex constraints, and it remains one of the state-of-the-art algorithms for LDA. For fair comparison, we use deterministic gradients for SGRLD.

We perform a grid search over the constant step-size for both algorithms, and we keep the best three for MLD and SGRLD. For each iteration, we build an empirical distribution by running 2,000,000 independent trials, and we compute its total variation with respect to the histogram generated by the true distribution.

Figure 1(a) reports the total variation error along the first dimension, where we can see that MLD outperforms SGRLD by a substantial margin. As dictated by our theory, all the MLD curves decay at the $O(T^{-1/2})$ rate until they saturate at the dicretization error level. In contrast, SGRLD lacks non-asymptotic guarantees, and there is no clear convergence rate we can infer from Figure 1(a).

The improvement along all other dimensions (i.e., topics with less observations) are even more significant; see **Appendix H.1**.

## 5.2  Latent Dirichlet Allocation with Wikipedia Corpus

An influential framework for topic modeling is the Latent Dirichlet Allocation (LDA) [3], which, given a text collection, requires to infer the posterior word distributions without knowing the exact topic for each word. The full model description is standard but somewhat convoluted; we refer to the classic [3] for details.

Each topic $k$ in LDA determines a word distribution $\boldsymbol{\pi}_k$, and suppose there are in total $K$ topics and $W + 1$ words. The variable of interest is therefore $\boldsymbol{\pi} := (\boldsymbol{\pi}_1, \boldsymbol{\pi}_2, ..., \boldsymbol{\pi}_K) \in \Delta_W \times \Delta_W \times \cdots \Delta_W$. Since this domain is a Cartesian product of simplices, we propose to use $\tilde{h}(\boldsymbol{\pi}) := \sum_{k=1}^K h(\boldsymbol{\pi}_k)$, where $h$ is the entropic mirror map (3.4), for SMLD. It is easy to see that all of our computations for Dirichlet posteriors generalize to this setting.

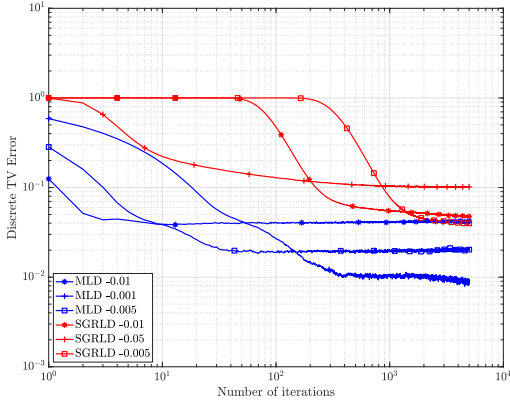
(a) Synthetic data, first dimension.

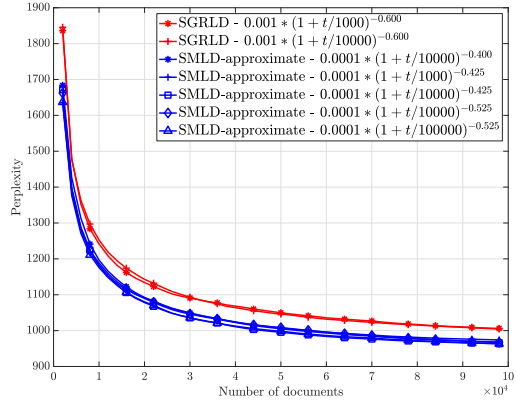
(b) LDA on Wikipedia corpus.

### 5.2.1 Experimental Setup

We implement the SMLD for LDA on the Wikipedia corpus with 100,000 documents, and we compare the performance against the SGRLD [26]. In order to keep the comparison fair, we adopt exactly the same setting as in [26], including the model parameters, the batch-size, the Gibbs sampler steps, etc. See Section 4 and 5 in [26] for omitted details.

Another state-of-the-art first-order algorithm for LDA is the SGRHMC in [21], for which we skip the implementation, due to not knowing how the $\hat{B}_t$ was chosen in [21]. Instead, we will repeat the same experimental setting as [21] and directly compare our results versus the ones reported in [21]. See **Appendix H.2** for comparison against SGRHMC.

### 5.2.2 A Numerical Trick and the SMLD-approximate Algorithm

A major drawback of the SMLD in practice is that the stochastic gradients (4.4) involve exponential functions, which are unstable for large-scale problems. For instance, in python, `np.exp(800) = inf`, whereas the relevant variable regime in this experiment extends to 1600. To resolve such numerical issues, we appeal to the linear approximation[4] $\exp(\mathbf{y}) \simeq \max\{0, 1 + \mathbf{y}\}$. Admittedly, our theory no longer holds under such numerical tricks, and we shall not claim that our algorithm is provably convergent for LDA. Instead, the contribution of MLD here is to identify the dual dynamics associated with (3.7), which would have been otherwise difficult to perceive. We name the resulting algorithm "SMLD-approximate" to indicate its heuristic nature.

### 5.2.3 Results

Figure 1(b) reports the perplexity on the test data up to 100,000 documents, with the five best step-sizes we found via grid search for SMLD-approximate. For SGRLD, we use the best step-sizes reported in [26].

From the figure, we can see a clear improvement, both in terms of convergence speed and the saturation level, of the SMLD-approximate over SGRLD. One plausible explanation for such phenomenon is that our MLD, as a simple unconstrained Langevin Dynamics, is less sensitive to discretization. On the other hand, the underlying dynamics for SGRLD is a more sophisticated Riemannian diffusion, which requires finer discretization than MLD to achieve the same level of approximation to the original continuous-time dynamics, and this is true even in the presence of noisy gradients and our numerical heuristics

**Acknowledgments**

This project has received funding from the European Research Council (ERC) under the European Union's Horizon 2020 research and innovation programme (grant agreement n° 725594 - time-data).

## Footnotes

[1] In general, (2.3) is ill-defined; see [30]. The validity of (2.3) is guaranteed by McCann's theorem [23] under **Assumption 2** and **3**.

[2]More examples of mirror map can be found in **Appendix B**.

[3]Our guarantee is given on a randomly chosen iterate from $\{\mathbf{x}^1, \mathbf{x}^2, ..., \mathbf{x}^T\}$, instead of the final iterate $\mathbf{x}^T$. In practice, we observe that the final iterate always gives the best performance, and we will ignore this minor difference in the theorem statement.

[4]One can also use a higher-order Taylor approximation for $\exp(\mathbf{y})$, or add a small threshold $\exp(\mathbf{y}) \simeq \max\{\epsilon, 1 + \mathbf{y}\}$ to prevent the iterates from going to the boundary. In practice, we observe that these variants do not make a huge impact on the performance.

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
