[Supplementary Material]

# Supplementary Materials for
# Mirrored Langevin Dynamics

**Ya-Ping Hsieh**      **Ali Kavis**      **Paul Rolland**      **Volkan Cevher**

Laboratory for Information and Inference Systems (LIONS),
EPFL, Lausanne, Switzerland
{ya-ping.hsieh, ali.kavis, paul.rolland, volkan.cevher}@epfl.ch

## A   Proof of Theorem 2

We first focus on the convergence for total variation and relative entropy, since they are in fact quite trivial. The proof for the 2-Wasserstein distance requires a bit more work.

### A.1   Total Variation and Relative Entropy

Since $h$ is strictly convex, $\nabla h$ is one-to-one, and hence

$$
\begin{aligned}
d_{\mathrm{TV}}(\nabla h \# \mu_1, \nabla h \# \mu_2) &= \frac{1}{2} \sup_E |\nabla h \# \mu_1(E) - \nabla h \# \mu_2(E)| \\
&= \frac{1}{2} \sup_E \left| \mu_1\big(\nabla h^{-1}(E)\big) - \mu_2\big(\nabla h^{-1}(E)\big) \right| \\
&= d_{\mathrm{TV}}(\mu_1, \mu_2).
\end{aligned}
$$

On the other hand, it is well-known that applying a one-to-one mapping to distributions leaves the relative entropy intact. Alternatively, we may also simply write (letting $\nu_i = \nabla h \# \mu_i$):

$$
\begin{aligned}
D(\nu_1 \| \nu_2) &= \int \log \frac{\mathrm{d}\nu_1}{\mathrm{d}\nu_2} \mathrm{d}\nu_1 \\
&= \int \log \left( \frac{\mathrm{d}\nu_1}{\mathrm{d}\nu_2} \circ \nabla h \right) \mathrm{d}\mu_1 && \text{by (A.5) below} \\
&= \int \log \frac{\mathrm{d}\mu_1}{\mathrm{d}\mu_2} \mathrm{d}\mu_1 && \text{by (2.1)} \\
&= D(\mu_1 \| \mu_2)
\end{aligned}
$$

The "in particular" part follows from noticing that $\mathbf{y}^t \sim \nabla h \# \mathbf{x}^t$ and $\mathbf{Y}_\infty \sim \nabla h \# \mathbf{X}_\infty$.

### A.2   2-Wasserstein Distance

Now, let $h$ be $\rho$-strongly convex. The most important ingredient of the proof is **Lemma 1** below, which is conceptually clean. Unfortunately, for the sake of rigor, we must deal with certain intricate regularity issues in the Optimal Transport theory. If the reader wishes, she/he can simply assume that the quantities (A.1) and (A.2) below are well-defined, which is always satisfied by any practical mirror map, and skip all the technical part about the well-definedness proof.

For the moment, assume $h \in \mathcal{C}^5$; the general case is given at the end. Every convex $h$ generates a Bregman divergence via $B_h(\mathbf{x}, \mathbf{x}') := h(\mathbf{x}) - h(\mathbf{x}') - \langle \nabla h(\mathbf{x}'), \mathbf{x} - \mathbf{x}' \rangle$. The

following key lemma allows us to relate guarantees in $\mathcal{W}_2$ between $\mathbf{x}^t$'s and $\mathbf{y}^t$'s. It can be seen as a generalization of the classical duality relation (A.4) in the space of probability measures.

**Lemma 1** (Duality of Wasserstein Distances)**.** *Let $\mu_1$, $\mu_2$ be probability measures satisfying* ***Assumptions 2*** *and* ***3***. *If $h$ is $\rho$-strongly convex and $\mathcal{C}^5$, then the* (A.1) *and* (A.2) *below are well-defined:*

$$\mathcal{W}_{B_h}(\mu_1, \mu_2) := \inf_{T:T\#\mu_1=\mu_2} \int B_h\left(\mathbf{x}, T(\mathbf{x})\right) \mathrm{d}\mu_1(\mathbf{x}) \tag{A.1}$$

*and (notice the exchange of inputs on the right-hand side)*

$$\mathcal{W}_{B_{h^\star}}(\nu_1, \nu_2) := \inf_{T:T\#\nu_1=\nu_2} \int B_{h^\star}\left(T(\mathbf{y}), \mathbf{y}\right) \mathrm{d}\nu_1(\mathbf{y}). \tag{A.2}$$

*Furthermore, we have*

$$\mathcal{W}_{B_h}(\mu_1, \mu_2) = \mathcal{W}_{B_{h^\star}}(\nabla h\#\mu_1, \nabla h\#\mu_2). \tag{A.3}$$

Before proving the lemma, let us see that the relation in $\mathcal{W}_2$ is a simple corollary of **Lemma 1**. Since $h$ is $\rho$-strongly convex, it is classical that, for any $\mathbf{x}$ and $\mathbf{x}'$,

$$\frac{\rho}{2}\|\mathbf{x} - \mathbf{x}'\|^2 \le B_h(\mathbf{x}, \mathbf{x}') = B_{h^\star}(\nabla h(\mathbf{x}'), \nabla h(\mathbf{x})) \le \frac{1}{2\rho}\|\nabla h(\mathbf{x}) - \nabla h(\mathbf{x}')\|^2. \tag{A.4}$$

Using **Lemma 1** and the fact that $\mathbf{y}^t \sim \nabla h\#\mathbf{x}^t$ and $\mathbf{Y}_\infty \sim \nabla h\#\mathbf{X}_\infty$, we conclude $\mathcal{W}_2(\mathbf{x}^t, \mathbf{X}_\infty) \le \frac{1}{\rho}\mathcal{W}_2(\mathbf{y}^t, \mathbf{X}_\infty)$. It hence remains to prove **Lemma 1** when $h \in \mathcal{C}^5$.

### A.2.1 Proof of Lemma 1 When $h \in \mathcal{C}^5$

We first prove that (A.2) is well-defined by verifying the sufficient conditions in **Theorem 3.6** of [7]. Specifically, we will verify **(C0)-(C2)** in p.554 of [7] when the transport cost is $B_{h^\star}$.

Since $h$ is $\rho$-strongly convex, $\nabla h$ is injective, and hence $\nabla h^\star = (\nabla h)^{-1}$ is also injective, which implies that $h^\star$ is strictly convex. On the other hand, the strong convexity of $h$ implies $\nabla^2 h^\star \preceq \frac{1}{\rho}I$, and hence $B_{h^\star}$ is globally upper bounded by a quadratic function.

We now show that the conditions **(C0)-(C2)** are satisfied. Since we have assumed $h \in \mathcal{C}^5$, we have $B_{h^\star} \in \mathcal{C}^4$. Since $B_{h^\star}$ is upper bounded by a quadratic function, the condition **(C0)** is trivially satisfied. On the other hand, since $h^\star$ is strictly convex, simple calculation reveals that, for any $\mathbf{y}'$, the mapping $\mathbf{y} \to \nabla_{\mathbf{y}'} B_{h^\star}(\mathbf{y}, \mathbf{y}')$ is injective, which is **(C1)**. Similarly, for any $\mathbf{y}$, the mapping $\mathbf{y}' \to \nabla_{\mathbf{y}} B_{h^\star}(\mathbf{y}, \mathbf{y}')$ is also injective, which is **(C2)**. By **Theorem 3.6** in [7], (A.2) is well-defined.

We now turn to (A.3), which will automatically establish the well-definedness of (A.1). We first need the following equivalent characterization of $\nabla h\#\mu = \nu$ [12]:

$$\int f\mathrm{d}\nu = \int f \circ \nabla h \mathrm{d}\mu \tag{A.5}$$

for all measurable $f$. Using (A.5) in the definition of $\mathcal{W}_{B_{h^\star}}$, we get

$$\mathcal{W}_{B_{h^\star}}(\nabla h\#\mu_1, \nabla h\#\mu_2) = \inf_T \int B_{h^\star}\left(T(\mathbf{y}), \mathbf{y}\right) \mathrm{d}\nabla h\#\mu_1(\mathbf{y})$$

$$= \inf_T \int B_{h^\star}\left((T \circ \nabla h)(\mathbf{x}), \nabla h(\mathbf{x})\right) \mathrm{d}\mu_1(\mathbf{x}),$$

where the infimum is over all $T$ such that $T\#(\nabla h\#\mu_1) = \nabla h\#\mu_2$. Using the classical duality $B_h(\mathbf{x}, \mathbf{x}') = B_{h^\star}(\nabla h(\mathbf{x}'), \nabla h(\mathbf{x}))$ and $\nabla h \circ \nabla h^\star(\mathbf{x}) = \mathbf{x}$, we may further write

$$\mathcal{W}_{B_{h^\star}}(\nabla h\#\mu_1, \nabla h\#\mu_2) = \inf_T \int B_h\left(\mathbf{x}, (\nabla h^\star \circ T \circ \nabla h)(\mathbf{x})\right) \mathrm{d}\mu_1(\mathbf{x}) \tag{A.6}$$

where the infimum is again over all $T$ such that $T\#(\nabla h\#\mu_1) = \nabla h\#\mu_2$. In view of (A.6), the proof would be complete if we can show that $T\#(\nabla h\#\mu_1) = \nabla h\#\mu_2$ if and only if $(\nabla h^\star \circ T \circ \nabla h)\#\mu_1 = \mu_2$.

For any two maps $T_1$ and $T_2$, we claim that

$$(T_1 \circ T_2)\#\mu = T_1\# (T_2\#\mu). \tag{A.7}$$

Indeed, for any Borel set $E$, we have, by definition of the push-forward,

$$(T_1 \circ T_2)\#\mu(E) = \mu\big((T_1 \circ T_2)^{-1}(E)\big)$$
$$= \mu\big((T_2^{-1} \circ T_1^{-1})(E)\big).$$

On the other hand, recursively applying the definition of push-forward to $T_1\# (T_2\#\mu)$ gives

$$T_1\# (T_2\#\mu) (E) = T_2\#\mu\big(T^{-1}(E)\big)$$
$$= \mu\big((T_2^{-1} \circ T_1^{-1})(E)\big)$$

which establishes (A.7).

Assume that $T\#(\nabla h\#\mu_1) = \nabla h\#\mu_2$. Then we have

$$\begin{aligned}(\nabla h^\star \circ T \circ \nabla h)\#\mu_1 &= \nabla h^\star\#(T\#(\nabla h\#\mu_1)) &&\text{by (A.7)}\\ &= \nabla h^\star\#(\nabla h\#\mu_2) &&\text{since } T\#(\nabla h\#\mu_1) = \nabla h\#\mu_2\\ &= (\nabla h^\star \circ \nabla h)\#\mu_2 &&\text{by (A.7) again}\\ &= \mu_2.\end{aligned}$$

On the other hand, if $(\nabla h^\star \circ T \circ \nabla h)\#\mu_1 = \mu_2$, then composing both sides by $\nabla h$ and using (A.7) yields $T\#(\nabla h\#\mu_1) = \nabla h\#\mu_2$, which finishes the proof.

### A.2.2 When $h$ is only $\mathcal{C}^2$

When $h$ is only $\mathcal{C}^2$, we will directly resort to (A.4). Let $T$ be any map such that $T\#(\nabla h\#\mu_1) = \nabla h\#\mu_2$, and consider the optimal transportation problem $\inf_T \int \|\mathbf{y} - T(\mathbf{y})\|^2 \mathrm{d}\nabla h\#\mu_1(\mathbf{y})$. By (A.4) and (A.5), we have

$$\inf_T \int \|\mathbf{y} - T(\mathbf{y})\|^2 \mathrm{d}\nabla h\#\mu_1(\mathbf{y}) = \inf_T \int \|\nabla h(\mathbf{x}) - (T \circ \nabla h)(\mathbf{x}))\|^2 \mathrm{d}\mu_1(\mathbf{x})$$
$$\geq \rho^2 \inf_T \int \|\mathbf{x} - (\nabla h^\star \circ T \circ \nabla h)(\mathbf{x}))\|^2 \mathrm{d}\mu_1(\mathbf{x})$$

where the infimum is over all $T$ such that $T\#(\nabla h\#\mu_1) = \nabla h\#\mu_2$. But as proven in **Appendix A.2.1**, this is equivalent to $(\nabla h^\star \circ T \circ \nabla h)\#\mu_1 = \mu_2$. The proof is finished by noting $\mathbf{y}^t \sim \nabla h\#\mathbf{x}^t$ and $\mathbf{Y}_\infty \sim \nabla h\#\mathbf{X}_\infty$.

## B  More Examples of Mirror Map and their Dual Distributions

In this section, we present more instances of mirror map other than on the simplex, and their corresponding dual distributions.

### B.1  Mirror map on the hypercube

On the hypercube $[-1, 1]^d$, a possible mirror map is

$$h(\mathbf{x}) = \frac{1}{2}\sum_{i=1}^{d}\Big((1 + x_i)\log(1 + x_i) + (1 - x_i)\log(1 - x_i)\Big).$$

It can easily be shown that

$$\frac{\partial h}{\partial x_i} = \operatorname{arctanh}(x_i), \qquad \frac{\partial^2 h}{\partial x_i \partial x_j} = \frac{\delta_{ij}}{1 - x_i^2}, \qquad \frac{\partial h^*}{\partial y_i} = \tanh(y_i)$$

which implies that $h$ is 1-strongly convex.

Since the Hessian matrix of $h$ is diagonal, we have:

$$\log \det \nabla^2 h(\mathbf{x}) = \sum_{i=1}^{d} \log \left( \frac{1}{1 - x_i^2} \right).$$

Then, using the definition of $W$, we obtain

$$W(\mathbf{y}) = V \circ \nabla h^*(\mathbf{y}) + \sum_{i=1}^{d} \log \left( \frac{1}{1 - \tanh^2(y_i)} \right)$$

$$= V \circ \nabla h^*(\mathbf{y}) + \sum_{i=1}^{d} \log \left( \frac{4}{\exp(2y_i) + 2 + \exp(-2y_i)} \right)$$

$$= V \circ \nabla h^*(\mathbf{y}) + \sum_{i=1}^{d} \log \left( \frac{1}{2}(1 + \cosh(2y_i)) \right).$$

## B.2  Mirror map on the Euclidean ball

On the unit ball $\{\mathbf{x} \in \mathbb{R}^d : \|\mathbf{x}\| \leq 1\}$, where $\|\cdot\|$ denote the Euclidean norm, a possible mirror map is
$$h(\mathbf{x}) = -\log(1 - \|\mathbf{x}\|) - \|\mathbf{x}\|.$$

We can compute:

$$\frac{\partial h}{\partial x_i} = \frac{x_i}{1 - \|\mathbf{x}\|}, \qquad \frac{\partial^2 h}{\partial x_i \partial x_j} = \frac{\delta_{ij}}{1 - \|\mathbf{x}\|} + \frac{x_i x_j}{\|\mathbf{x}\|(1 - \|\mathbf{x}\|)^2}, \qquad \frac{\partial h^*}{\partial y_i} = \frac{y_i}{1 + \|\mathbf{y}\|}.$$

The Hessian matrix can thus be written as $\nabla^2 h(\mathbf{x}) = \frac{1}{1 - \|\mathbf{x}\|} I + \frac{1}{\|\mathbf{x}\|(1 - \|\mathbf{x}\|)^2} \mathbf{x}\mathbf{x}^T$, where $I$ is the identity matrix. Invoking the matrix determinant lemma, we get

$$\det(\nabla^2 h(\mathbf{x})) = \left( 1 + \frac{\mathbf{x}^\top \mathbf{x}}{\|\mathbf{x}\|(1 - \|\mathbf{x}\|)} \right) \det \left( \frac{1}{1 - \|\mathbf{x}\|} I \right) = \left( \frac{1}{1 - \|\mathbf{x}\|} \right)^{d+1}.$$

We thus obtain:

$$W(\mathbf{y}) = V \circ \nabla h^*(\mathbf{y}) - (d + 1) \log \left( 1 - \frac{\|\mathbf{y}\|}{1 + \|\mathbf{y}\|} \right)$$

$$= V \circ \nabla h^*(\mathbf{y}) + (d + 1) \log \left( 1 + \|\mathbf{y}\| \right).$$

# C   Proof of Thereom 3

In previous sections, we are given a target distribution $e^{-V}$ and a mirror map $h$, and we derive the induced distribution $e^{-W}$ through the Monge-Ampère equation (2.1). The high-level idea of this proof is to reverse the direction: We start with two good distributions $e^{-V}$ and $e^{-W}$, and we invoke deep results in Optimal Transport to deduce the existence of a good mirror map $h$.

First, notice that if $V$ has bounded domain, then the strong convexity of $V$ implies $V > -\infty$. Along with the assumption that $V$ is bounded away from $+\infty$ in the interior, we see that $e^{-V}$ is bounded away from 0 and $+\infty$ in the interior of support.

Let $d\nu(\mathbf{x}) = e^{-W(\mathbf{x})} d\mathbf{x}$ be any distribution such that $\nabla^2 W \succeq I$. By Brenier's polarization theorem [1, 2] and **Assumption 2, 3**, there exists a convex function $h^\star$ whose gradient solves the $\mathcal{W}_2(\nu, \mu)$ optimal transportation problem. Caffarelli's regularity theorem [3–5] then implies that the Brenier's map $h^\star$ is in $\mathcal{C}^2$. Finally, a slightly stronger form of Caffarelli's contraction theorem [10] asserts:

$$\nabla^2 h^\star \preceq \frac{1}{m} I, \tag{C.1}$$

which implies $h = (h^\star)^\star$ is $m$-strongly convex.

Let us consider the discretized MLD (3.3) corresponding to the mirror map $h$. Invoking **Theorem 3** of [6], the convergence rate of the discretized Langevin dynamics $\mathbf{y}^T$ for $\mu$ is such that $D(\mathbf{y}^T \| \nu) = \tilde{O}(d/T)$, which in turn implies $\mathcal{W}_2(\mathbf{y}^T, \nu) = \tilde{O}\left(\sqrt{d/T}\right)$ and $d_{\mathrm{TV}}(\mathbf{y}^T, \nu) = \tilde{O}\left(\sqrt{d/T}\right)$. **Theorem 2** then completes the proof.

# D Proof of Lemma 1

Straightforward calculations in convex analysis shows

$$\frac{\partial h}{\partial x_i} = \log \frac{x_i}{x_{d+1}}, \qquad \frac{\partial^2 h}{\partial x_i \partial x_j} = \delta_{ij} x_i^{-1} + x_{d+1}^{-1},$$

$$h^\star(\mathbf{y}) = \log\left(1 + \sum_{i=1}^d e^{y_i}\right), \quad \frac{\partial h^\star}{\partial y_i} = \frac{e^{y_i}}{1 + \sum_{i=1}^d e^{y_i}}, \tag{D.1}$$

which proves that $h$ is 1-strongly convex.

Let $\mu = e^{-V(\mathbf{x})} d\mathbf{x}$ be the target distribution and define $\nu = e^{-W(\mathbf{y})} d\mathbf{y} := \nabla h \# \mu$. By (2.1), we have

$$W \circ \nabla h = V + \log \det \nabla^2 h. \tag{D.2}$$

Since $\nabla^2 h(\mathbf{x}) = \mathrm{diag}[x_i^{-1}] + x_{d+1}^{-1} \mathbb{1}\mathbb{1}^\top$ where $\mathbb{1}$ is the all 1 vector, the well-known matrix determinant lemma "$\det(A + \mathbf{u}\mathbf{v}^\top) = (1 + \mathbf{v}^\top A^{-1} \mathbf{u}) \det A$" gives

$$\log \det \nabla^2 h(\mathbf{x}) = \log\left(1 + x_{d+1}^{-1} \sum_{i=1}^d x_i\right) \cdot \prod_{i=1}^d x_i^{-1}$$

$$= -\sum_{i=1}^{d+1} \log x_i = -\sum_{i=1}^d \log x_i - \log\left(1 - \sum_{i=1}^d x_i\right). \tag{D.3}$$

Composing both sides of (D.2) with $\nabla h^\star$ and using (D.1), (D.3), we then finish the proof by computing

$$W(\mathbf{y}) = V \circ \nabla h^\star(\mathbf{y}) - \sum_{i=1}^d y_i + (d+1)\log\left(1 + \sum_{i=1}^d e^{y_i}\right)$$

$$= V \circ \nabla h^\star(\mathbf{y}) - \sum_{i=1}^d y_i + (d+1)h^\star(\mathbf{y}).$$

# E Proof of Lemma 2

The proof relies on rather straightforward computations.

1. In order to show $e^{-(W+C)} = \nabla h \# e^{-V}$ for some constant $C$, we will verify the Monge-Ampère equation:

$$e^{-V} = e^{-(W \circ \nabla h + C)} \det \nabla^2 h \tag{E.1}$$

for $V = \sum_{i=1}^N V_i$ and $W = \sum_{i=1}^N W_i$, where $W_i$ is defined via (4.2). By (4.2), it holds that

$$\frac{1}{C_i} e^{-NV_i} = e^{-NW_i \circ \nabla h} \det \nabla^2 h, \quad C_i := \frac{1}{\int e^{-NV_i}}. \tag{E.2}$$

Multiplying (E.2) for $i = 1, 2, ..., N$, we get

$$\prod_{i=1}^N \frac{1}{C_i} e^{-NV} = e^{-NW \circ \nabla h} \left(\det \nabla^2 h\right)^N. \tag{E.3}$$

The first claim follows by taking the $N^{\mathrm{th}}$ root of (E.3).

2. The second claim directly follows by (E.2).

3. Trivial.

4. By (E.1) and (E.2) and using $\nabla h^\star \circ \nabla h(\mathbf{x}) = \mathbf{x}$, we get

$$W_i = V_i \circ \nabla h^\star + \frac{1}{N} \log \det \nabla^2 h(\nabla h^\star) - \log C_i, \tag{E.4}$$

$$W = V \circ \nabla h^\star + \log \det \nabla^2 h(\nabla h^\star) - C, \tag{E.5}$$

which implies $N\nabla W_i - \nabla W = \nabla^2 h^\star \left( N\nabla V_i \circ \nabla h^\star - \nabla V \circ \nabla h^\star \right)$. Since $h$ is 1-strongly convex, $h^\star$ is 1-Lipschitz gradient, and therefore the spectral norm of $\nabla^2 h^\star$ is upper bounded by 1. In the case of $b = 1$, the final claim follows by noticing

$$\mathbb{E}\|\tilde{\nabla} W - \nabla W\|^2 = \frac{1}{N} \sum_{i=1}^{N} \|N\nabla W_i - \nabla W\|^2 \tag{E.6}$$

$$= \frac{1}{N} \sum_{i=1}^{N} \|\nabla^2 h^\star \left( N\nabla V_i \circ \nabla h^\star - \nabla V \circ \nabla h^\star \right)\|^2 \tag{E.7}$$

$$\leq \frac{\|\nabla^2 h^\star\|_{\text{spec}}^2}{N} \sum_{i=1}^{N} \|N\nabla V_i \circ \nabla h^\star - \nabla V \circ \nabla h^\star\|^2 \tag{E.8}$$

$$\leq \mathbb{E}\|\tilde{\nabla} V - \nabla V\|^2. \tag{E.9}$$

The proof for general batch-size $b$ is exactly the same, albeit with more cumbersome notation.

# F  Proof of Theorem 4

The proof is a simple combination of the existing result in [8] and our theory in Section 3.

By **Theorem 2**, we only need to prove that the inequality (4.3) holds for $D(\tilde{\mathbf{y}}^T \| e^{-W(\mathbf{y})} d\mathbf{y})$, where $\tilde{\mathbf{y}}^T$ is to be defined below. By assumption, $W$ is unconstrained and satisfies $LI \succeq \nabla^2 W \succeq 0$. By **Lemma 2**, the stochastic gradient $\tilde{\nabla} W$ is unbiased and satisfies

$$\mathbb{E}\|\tilde{\nabla} W - \nabla W\|^2 \leq \mathbb{E}\|\tilde{\nabla} V - \nabla V\|^2 = \sigma^2.$$

Pick a random index[1] $t \in \{1, 2, ..., T\}$ and set $\tilde{\mathbf{y}}^T := \mathbf{y}^t$. Then **Corollary 18** of [8] with $D^2 = \sigma^2$ and $M_2 = 0$ implies $D(\tilde{\mathbf{y}}^T \| e^{-W(\mathbf{y})} d\mathbf{y}) \leq \epsilon$, provided

$$\beta \leq \min\left\{ \frac{\epsilon}{2(Ld + \sigma^2)}, \frac{1}{L} \right\}, \quad T \geq \frac{\mathcal{W}_2^2(\mathbf{y}^0, e^{-W(\mathbf{y})} d\mathbf{y})}{\beta\epsilon}. \tag{F.1}$$

Solving for $T$ in terms of $\epsilon$ establishes the theorem.

# G  Stochastic Gradients for Dirichlet Posteriors

In order to apply SMLD, one must have, for each term $V_i$, the corresponding dual $W_i$ defined via (4.2). In this appendix, we derive a closed-form expression in the case of the Dirichlet posterior (3.6).

Recall that the Dirichlet posterior (3.6) consists of a Dirichlet prior and categorical data observations [9]. Let $N := \sum_{\ell=1}^{d+1} n_\ell$, where $n_\ell$ is the number of observations for category $\ell$, and suppose that the parameters $\alpha_\ell$'s are given. If the $i^{\text{th}}$ data is in category $c_i \in \{1, 2, ..., d+1\}$, then we can define $V_i(\mathbf{x}) := -\sum_{\ell=1}^{d+1} \mathbb{I}_{\{\ell=c_i\}} \log x_\ell - \frac{1}{N} \sum_{\ell=1}^{d+1} (\alpha_\ell - 1) \log x_\ell$

(a) Synthetic data, $8^{\text{th}}$ dimension.　　　　　(b) LDA on Wikipedia corpus.

Figure 1: LDA for Wikipedia, 50 topics.

so that **Assumption 4** holds. In view of **Lemma 1**, The corresponding dual $W_i$ is, up to a constant, given by

$$W_i(\mathbf{y}) = -\sum_{\ell=1}^{d} \mathbb{I}_{\{\ell=c_i\}} y_\ell - \sum_{\ell=1}^{d} \frac{\alpha_\ell}{N} y_\ell + h^\star + \left(\sum_{\ell=1}^{d+1} \frac{\alpha_\ell}{N}\right) h^\star(\mathbf{y}). \tag{G.1}$$

Similarly, if we take a mini-batch $B$ of the data with $|B| = b$, then

$$\frac{N}{b}\tilde{W}(\mathbf{y}) \coloneqq \frac{N}{b} \sum_{i \in B} W_i(\mathbf{y}) = -\sum_{\ell=1}^{d} \left(\frac{Nm_\ell}{b} + \alpha_\ell\right) y_\ell + \left(N + \sum_{\ell=1}^{d+1} \alpha_\ell\right) h^\star(\mathbf{y}), \tag{G.2}$$

where $m_\ell$ is the number of observations of category $\ell$ in the set $B$. Apparently, the gradient of (G.2) is (4.4).

## H　More on Experiments

### H.1　Synthetic Data

Figure 1(a) reports the total variation error along the $8^{\text{th}}$ dimension of the synthetic experiment in Section 5.1. Compared to Figure 1(a) in the main text, it is evident that MLD achieves an even stronger performance than SGRLD, especially in the saturation error phase.

### H.2　Comparison against SGRHMC for Latent Dirichlet Allocation

The only difference between the experimental setting of [11] and the main text is the number of topics (50 vs. 100). In this appendix, we run SMLD-approximate under the setting of [11] and directly compare against the results reported in [11]. We have also included the SGRLD as a baseline.

Figure 1(b) reports the perplexity on the test data. According to [11], the best perplexity achieved by SGRHMC up to 10,000 documents is approximately 1400, which is worse than the 1323 by SMLD-approximate. Moreover, from Figure 3 of [11], we see that the SGRHMC yields comparable performance as SGRLD for 2 out 3 independent runs, especially in the beginning phase, whereas the SMLD-approximate has sizeable lead over SGRLD at any stage of the experiment. The potential reason for this improvement is, similar to SGRLD, that the SGRHMC exploits the Riemannian Hamiltonian dynamics, which is more complicated than MLD and hence more sensitive to the discretization error.

## Footnotes

[1]The analysis in [8] provides guarantees on the probability measure $\nu_T := \frac{1}{N} \sum_{t=1}^{T} \nu_t$ where $\mathbf{y}^t \sim \nu_t$. The $\tilde{\mathbf{y}}^T$ defined here has law $\nu_T$.