[Reviews · NeurIPS 2018]

Reviewer 1



The paper proposes to sample from constrained distributions by first mapping the original space into an unconstrained space and then using a Langevin sampler. When the potential in the new space is strongly convex, this approach gives a convergence speed for constrained spaces similar to that of unconstrained spaces. The method is illustrated in Dirichlet and LDA models. The paper is nicely written and the results interesting. A concern I have is that it is not clear which other constrained distributions admit a mirror map with an explicit strongly convex function W(y). For example, the important case of truncated (multivariate) Gaussians is not mentioned.

Reviewer 2



This is a very well-written paper and excellently presented with some interesting supporting theoretical results. The paper introduces a method (mirror map) from the optimization literature, mirrored descent, to perform scalable Monte Carlo sampling in a constrained state space. The mirror map works by transforming the sampling problem onto an unconstrained space, where stochastic gradient Markov chain Monte Carlo (MCMC) algorithms, in particular, stochastic gradient Langevin dynamics, can be readily applied. The Fenchal dual of the transformation function is used to transform the samples from the unconstrained space back onto the constrained space. In the paper, the authors state that a "good" mirror map is required. It isn't quite clear what this means. It would seem from the main paper that the mirror map function, h, should be twice continuously differentiable. However, this seems to conflict with the results in the appendix where h \in C^5 and not h \in C^2. Could the authors please clarify? In Section 3.2.2. (simplex section), a mirror map is given for the simplex example, however, it isn't clear how this is derived, or in fact, how one would create their own mirror in a different setting, e.g. sampling on the positive real line. Is there a general recipe that can be applied? How do you know that the chosen mirror map is optimal, assuming that there exists a wider class of mirror maps? My main criticism of the paper is related to the simulation study, which is quite limited. It's a little strange that only two Dirichlet examples are given even though the authors could apply their methodology to other constrained spaces. Is there a reason why only the Dirichlet was considered? Figure 1 only plots the results for the first dimension. Can we assume that the results are the same for all dimensions? This is an important point as it isn't completely clear whether or not the proposed approach works better than Patterson & Teh method (SGRLD) when sampling near to the boundary. It also appears from the simulation study that the improvement offered by the mirror map is only minor compared to SGRLD, could the authors comment on this further? Perhaps explain whether or not there are scenarios where SGRLD would be better. Overall, this is a nice contribution to the stochastic gradient MCMC literature with strong supporting theoretical results. A particularly nice outcome of this paper is related to the simplex example, where the mirror map approach transforms the sampling problem from being non-log-concave to strictly log-concave.

Reviewer 3



This paper deals with the simulation of a distribution whose support is constrained to a set. This is an important problem that had not received a really satisfactory answer (the best known "generic" algorithms having dimensional dependencies making them quite unsuitable for large-dimensional model inference). This contribution uses the so-called mirror maps. The mirror maps may be used to transform a constrained problem into an unconstrained one, for which the standard sampling algorithm may be used. The graal is to find a transformation of the variables which is such that the resulting sampling problem is both unconstrained and convex or strongly convex. This will not, in general, be the case, but in the case where such property is satisfied, then the resulting algorithm directly inherits the nice mixing property of (strongly) convex optimization over the unconstrained domain (allowing to apply the SGLD or other variants). This might lead to an algorithm whose mixing rate scales linearly in the dimension. This is in particularly true for sampling the posterior distribution over the probability simplex. This is a very important example, with many potential applications. On the LDA, the proposed solution outperforms the state-of-the art Minor comments: - Clearly, the paper was written very quickly and contains a very large number of typos, in some cases several in the same line "theif rates are significantly worse than the unconstrained scenaria under the same assumtions" - Theorem 1: the results are already stated (without a proof) in Section 2.2 (I have perhaps missed something)